# Experimental Study on a New Combined Gas–Liquid Separator

**Lei Ji, Qin Zhao \*, Huiming Deng, Lanyue Zhang and Wanquan Deng**

Key Laboratory of Fluid and Power Machinery, Ministry of Education, Xihua University, Chengdu 610039, China; 0119910030@mail.xhu.edu.cn (L.J.); denghuiming@stu.xhu.edu.cn (H.D.); 212019080700005@stu.xhu.edu.cn (L.Z.); 0119960034@mail.xhu.edu.cn (W.D.)

\* Correspondence: zhaoqin@mail.xhu.edu.cn

**Abstract:** Gas–liquid separation at natural gas wellheads has always been a key technical problem in the fields of natural gas transportation and storage. Developing a gas–liquid separation device that is both universal and highly efficient is the current challenge. A new type of combined gas–liquid separation device was designed in this study, and the efficiency of the separator was studied using a laser Doppler anemometer and phase Doppler particle analyzer at a flow rate of 10–60 Nm$^3$/h. The results showed that the separation efficiency of the combined separator was above 95% at each experimental flow rate, verifying the strong applicability of the combined separator. Moreover, the separation efficiency was as high as 99% at the flow rates of 10 and 60 Nm$^3$/h, thereby realizing efficient separation. This study is significant to the development of gas–liquid separation devices.

**Keywords:** natural gas; gas–liquid separation device; laser Doppler anemometer; phase Doppler particle analyzer



## 1. Introduction

With the rapid development of the global economy and industries, the increasing demand for energy and worsening environmental issues have become major concerns. Natural gas has received attention as a clean and efficient high-quality energy source, and its development and application have made significant advances [1,2]. In the industrial context of increased demand for natural gas, transportation and storage concerns have gained significant attention.

The natural gas coming out of the wellhead contains saturated water vapor and a small amount of hydrocarbons. Inside the pipeline, some components of natural gas react with free water, which leads to the formation of acidic substances and hydrate crystallization; this, in turn, causes corrosion and pipeline blockages that reduce the gas transmission efficiency, thus increasing gas supply instability and economic losses [3,4]. Removal of free water before the natural gas enters the pipeline has become a key issue in the subsequent stages of natural gas transportation [5,6]. At present, the traditional natural gas dehydration methods used are the solid adsorption, solvent absorption and low temperature condensation methods [7].

The solid adsorption method mainly involves the use of molecular sieves, silica gel, alumina and other substances with good absorption capacity for water as adsorbents. When natural gas is in contact with such substances, the water in the gas is absorbed to achieve a dehydration effect. However, this method has disadvantages, such as the associated high cost and the requirement of a large space [8]. The solvent absorption method mainly involves the use of absorbents with a high absorption capacity for water, and dehydration equipment can be designed according to the absorbent capacity of the absorbents. Triethylene glycol is the most widely used absorbent for achieving the required dehydration effect. However, triethylene glycol dehydration equipment has several components, high complexity, high maintenance cost and risk of environmental pollution with improper operation [9]. In the low-temperature condensation method, the temperature of natural gas

is reduced through throttling expansion and external refrigerant heat exchange such that the water in it condenses out, thereby achieving gas–liquid separation. However, throttling expansion refrigeration is only suitable for high-pressure aqueous natural gas, which also has limitations, while the external refrigerant heat exchange refrigeration method uses complex devices and consumes a large amount of energy [10]. Therefore, traditional natural gas dehydration methods are limited. With the continuous development of technologies, various researchers have optimized and improved the traditional dehydration methods of natural gas and gradually derived new natural gas dehydration methods, such as the ionic liquid natural gas dehydration method [11] and supersonic separator dehydration method [12]. These various methods have advantages and disadvantages, are not versatile and the separation mechanism of most separation equipment is not yet clear [13]. Therefore, the investigation, development and fabrication of efficient, low-resistance, universal natural gas dehydration methods and equipment is important in the field of natural gas transportation and storage. Compared with other gas-liquid separators, cyclone separators have the advantages of small size and long maintenance intervals; thus, they have become the focus of research. Matsubayashi et al. [14] studied an aero-liquid cyclone separator in a boiling water reactor and concluded that the diameter of the hub does not affect the efficiency of the separator; however, reducing the leaf placement angle will reduce the separation efficiency. Yu et al. [15] studied the effect of droplet particle size on the efficiency of a cyclone separator using Fluent software, and concluded that as the droplet particle size increases, the separator efficiency increases, and when the particle size is greater than 10 µm, the separation efficiency remains fixed. Han et al. [16] applied numerical calculations to study the flow field pressure, velocity and separation efficiency of a cyclone separator and concluded that with an increase in the blade envelope arc, the pressure, speed and separation efficiency of the internal flow field fluctuated greatly; they proposed that the blade envelope should be set in the range of 32–44 mm.

However, although several studies have been conducted on cyclone–liquid separators, the problem of low efficiency of a cyclone separator under small flow conditions is still not resolved. To improve the efficiency and applicability of the gas–liquid separator, this study optimized a design based on the cyclone separator. In this study, a combined natural gas dewatering device was designed, and the dewatering effect of the combined separator was measured using a laser Doppler anemometer (LDA) and phase Doppler particle analyzer (PDPA). This study has great research significance in the fields of natural gas transportation, storage and gas–liquid separation.

## 2. Experimental Principle and Process

The components of the combined separator designed in this study include the cyclone, steady flow, leaf grid and folding plate elements, as shown in Figure 1. The cyclone element is a preliminary separation component that relies on twisted blades to form a swirling flow of gas–liquid mixed fluid and the centrifugal action is applied to achieve the effect of gas–liquid separation [17]. The structure is similar to the internal flow field of a hydraulic mechanical transfer wheel chamber; after the fluid passes through the twisted blade, the flow state is twisted from the layer flow to turbulence, and the flow is extremely unstable [18,19]. This unstable flow can negatively impact the subsequent flow measurement if left untreated [20,21]. Therefore, a steady flow element is added downstream of the cyclone element to smooth the flow state of the strong turbulent fluid and have a certain separation effect on the droplets. Downstream of the steady flow element, the leaf grid and folding plate elements are installed sequentially and separated from the liquid contained in the natural gas to improve the separation efficiency according to the principle of inertial separation.

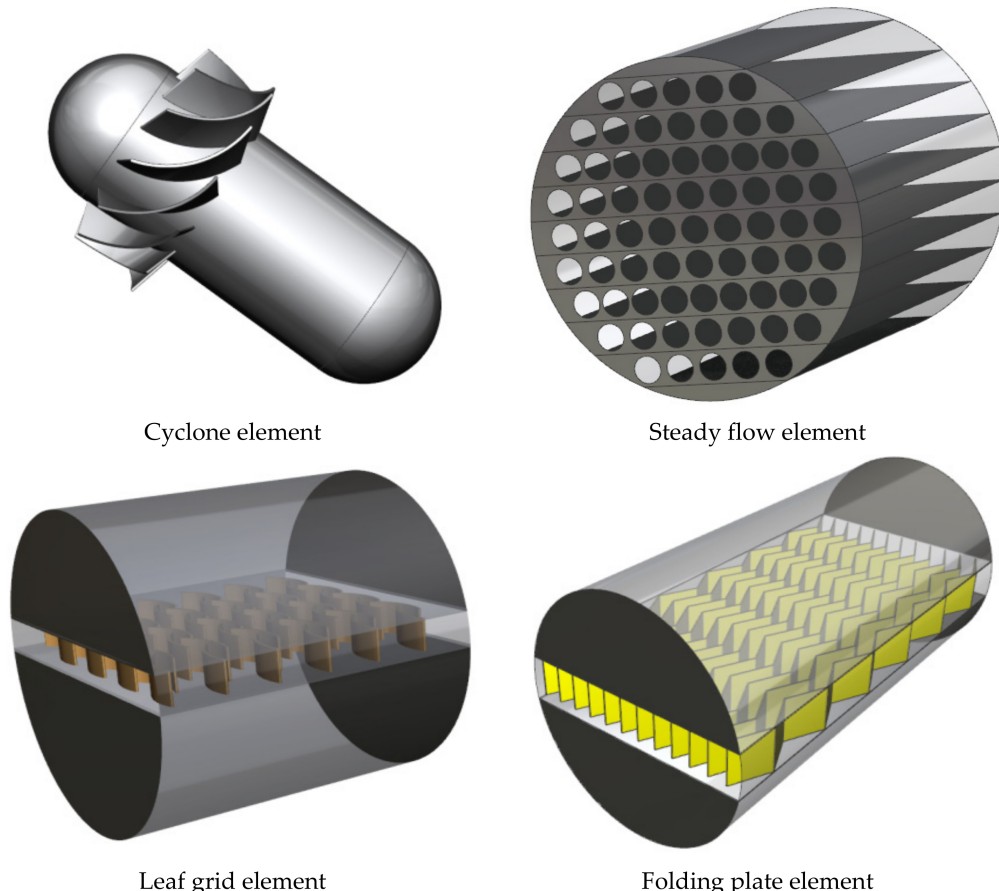

|  |  |
|---|---|
| Cyclone element | Steady flow element |
| Leaf grid element | Folding plate element |

**Figure 1.** Schematic of the separate components.

LDA is an application of the optical Doppler effect that relies on the frequency difference between the scattered light of moving particles and the irradiated light to obtain speed information and determine the particle size by analyzing the phase difference of the scattered light reflected or refracted by spherical particles passing through the measuring body of the laser [22,23]. Due to its laser measurement, the lack of interference from the convection field, wide range of velocity measurements and high accuracy, [24,25], it is now widely used in fluid flow rate measurements [26–29]. In this study, LDA and PDPA equipment were used to measure the liquid content in natural gas, and the equipment model is listed in Table 1.

**Table 1.** Basic device parameters.

| Name | Type Specification |
|---|---|
| Integrated argon-ion laser | LA70-5 |
| Beam splitter | FBL-3 fiberlight$^{TM}$ |
| 2D fiber optic emission probe | TM250 |
| Fiber optic receive probe | RV3070 PDPA |
| Three-channel photodetector assembly | PDM1000-3P |

To ensure the safety of the experiment, air was used as the experimental gas. Figure 2 presents a schematic of the experimental platform. The air is compressed by the air compressor (Figure 3) and enters the experimental apparatus from the inlet pipeline. Figure 4 shows the atomization system, which atomizes the liquid water and injects it into the experimental pipeline through the nozzle to mix with the air and form a gas–liquid two-phase flow. The atomized water particle size ($W_{1d}$) and atomized water content ($W_{1c}$) in the gas–liquid two-phase flow at monitoring point 1 ($M_1$) are measured using LDA and

PDPA devices. After performing a series of gas–liquid separation processes through the cyclone, steady flow, leaf grid and folding plate elements, the LDA and PDPA devices were used at monitoring point 2 ($M_2$) to measure the atomized water particle size ($W_{2d}$) and content ($W_{2c}$) in the gas–liquid two-phase flow through the combined separator. The separation efficiency was calculated using Equation (1). Figure 5 presents an image of the field experimental site.

$$\eta = \frac{W_{1c} - W_{2c}}{W_{1c}} \tag{1}$$

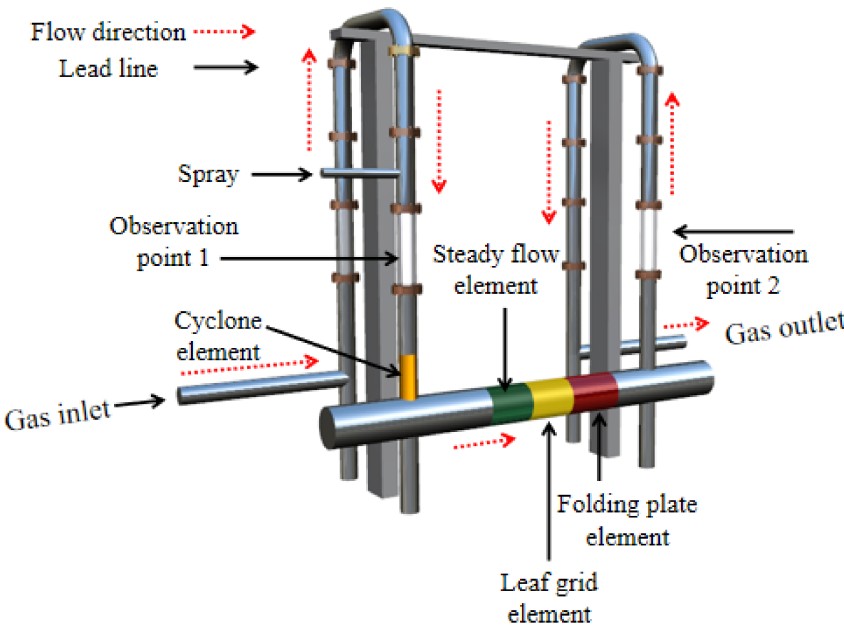

**Figure 2.** Experimental schematic.

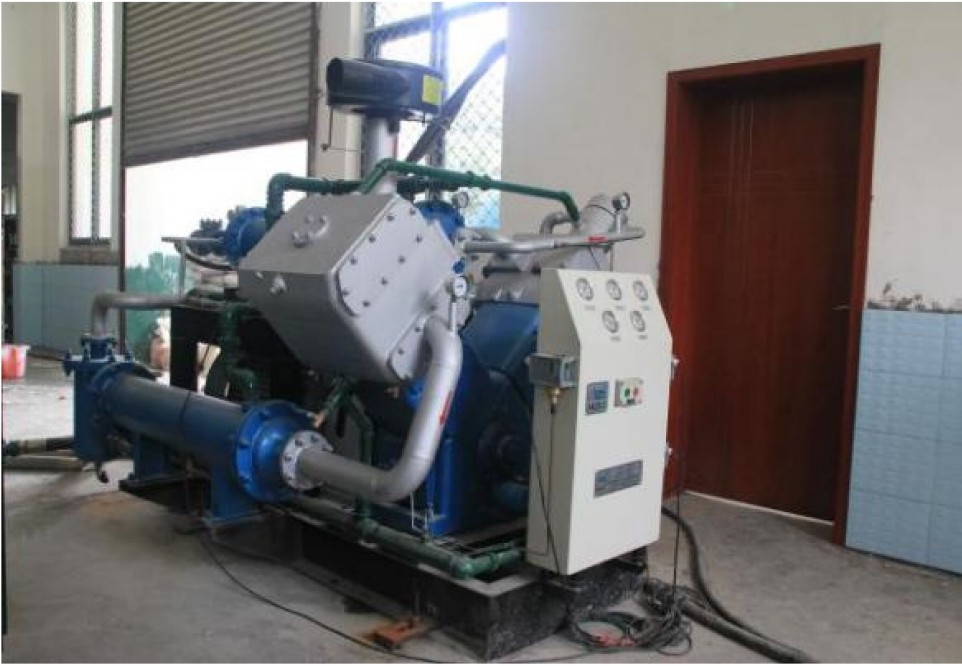

**Figure 3.** Air compressor.

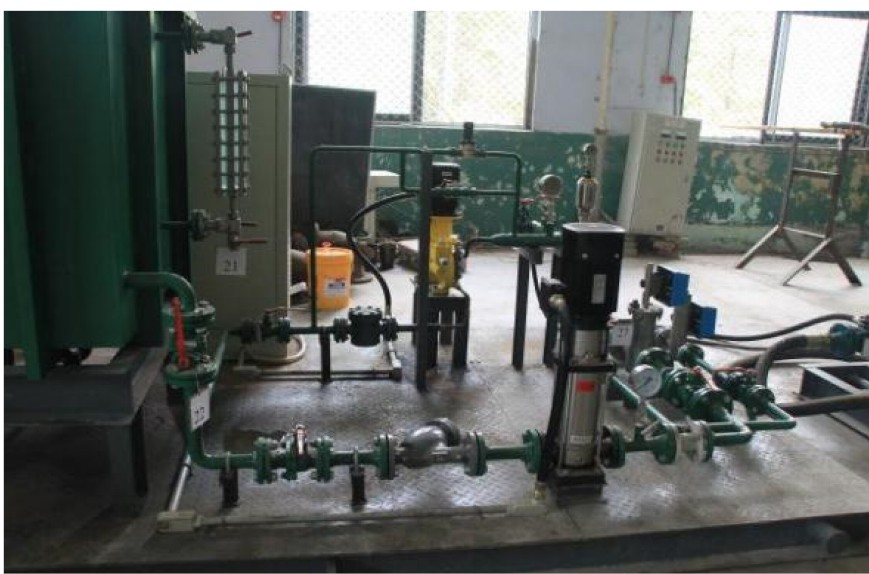

**Figure 4.** Atomization generator.

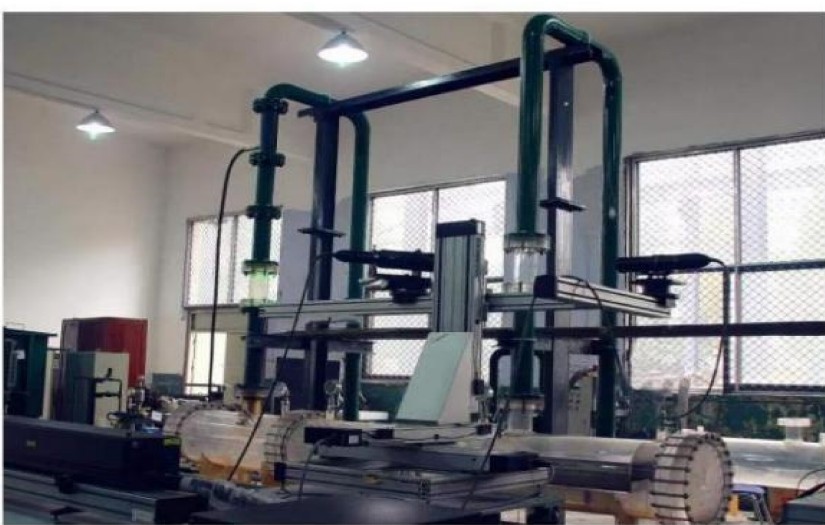

**Figure 5.** Experimental site.

During the experiment, the pipeline pressure was maintained at 0.055 MPa, and the pipeline temperature was 20 °C. Four sets of comparative cases were designed to explore the influence of each element on the liquid separation effect under different flow conditions, as shown in Table 2. Each case was measured in six flow conditions (10, 20, 30, 40, 50 and 60 Nm³/h). During the preliminary experiments, we found that if the spray flow is large, the liquid will form a femoral liquid, and the separation efficiency of various cases in this state is larger, reducing the significance of the study and deviating from the actual working conditions. After repeated debugging, the optimal spray flow rate was determined as 10.44 L/h. At this flow rate, the liquid forms a mist.

**Table 2.** Experimental case table.

| Case | Cyclone Element | Steady Flow Element | Leaf Grid Element | Folding Plate Element |
|------|-----------------|---------------------|-------------------|-----------------------|
| 1 | √ | | | |
| 2 | √ | | √ | |
| 3 | √ | √ | √ | |
| 4 | √ | √ | √ | √ |

### 3. Experimental Results and Analysis

*3.1. Study on the Separation Effect of Cyclone Elements*

To reduce the number of experimental errors, four replicates of the experiment per operating condition were conducted, and the average of the four experiments was taken as the result. Tables 3–6 list the measurement results of the liquid particle size (effective particle size D10) and the liquid content measurements at $M_1$ and $M_2$ under different flow conditions in Case 1. Figure 6 presents the $M_1$ and $M_2$ liquid particle size and content comparative analysis chart. As can be seen in the figure, the impact of the air flow on the atomized water increased as the air flow increased, affecting the diffusion of the atomized water. An increase in air flow led to an increase in the droplet particle size at $M_1$. For example, the particle size at $M_1$ was 21.15 µm at the flow rate of 10 $Nm^3/h$, and the particle size at $M_1$ was 24.85 µm at 60 $Nm^3/h$ (particle size increase of 3.7 µm).

**Table 3.** Liquid particle size in Case 1 at $M_1$ (µm).

| Number of Experiments | 10 $Nm^3/h$ | 20 $Nm^3/h$ | 30 $Nm^3/h$ | 40 $Nm^3/h$ | 50 $Nm^3/h$ | 60 $Nm^3/h$ |
|---|---|---|---|---|---|---|
| 1 | 21.03 | 21.51 | 21.63 | 22.67 | 23.00 | 24.90 |
| 2 | 21.15 | 21.36 | 21.96 | 22.82 | 23.34 | 24.78 |
| 3 | 21.30 | 21.48 | 21.9 | 22.4 | 23.15 | 24.84 |
| 4 | 21.13 | 21.25 | 22.02 | 22.58 | 23.42 | 24.89 |
| Average value | 21.15 | 21.40 | 21.88 | 22.62 | 23.23 | 24.85 |

**Table 4.** Liquid content in Case 1 at $M_1$ ($g/m^3$).

| Number of Experiments | 10 $Nm^3/h$ | 20 $Nm^3/h$ | 30 $Nm^3/h$ | 40 $Nm^3/h$ | 50 $Nm^3/h$ | 60 $Nm^3/h$ |
|---|---|---|---|---|---|---|
| 1 | 957.94 | 406.47 | 255.08 | 215.31 | 155.55 | 133.55 |
| 2 | 960.54 | 435.11 | 249.36 | 225.74 | 164.27 | 128.68 |
| 3 | 955.60 | 427.70 | 265.77 | 205.05 | 152.68 | 125.44 |
| 4 | 948.91 | 419.74 | 249.82 | 218.85 | 159.71 | 123.41 |
| Average value | 955.75 | 422.26 | 255.01 | 216.24 | 158.05 | 127.77 |

**Table 5.** Liquid particle size in Case 1 at $M_2$ (µm).

| Number of Experiments | 10 $Nm^3/h$ | 20 $Nm^3/h$ | 30 $Nm^3/h$ | 40 $Nm^3/h$ | 50 $Nm^3/h$ | 60 $Nm^3/h$ |
|---|---|---|---|---|---|---|
| 1 | 14.00 | 14.47 | 14.41 | 14.55 | 14.34 | 13.92 |
| 2 | 13.95 | 14.57 | 14.51 | 14.41 | 14.36 | 13.92 |
| 3 | 14.14 | 14.46 | 14.44 | 14.51 | 14.37 | 13.87 |
| 4 | 14.01 | 14.44 | 14.47 | 14.42 | 14.36 | 13.99 |
| Average value | 14.03 | 14.49 | 14.46 | 14.47 | 14.36 | 13.93 |

**Table 6.** Liquid content in Case 1 at $M_2$ ($g/m^3$).

| Number of Experiments | 10 $Nm^3/h$ | 20 $Nm^3/h$ | 30 $Nm^3/h$ | 40 $Nm^3/h$ | 50 $Nm^3/h$ | 60 $Nm^3/h$ |
|---|---|---|---|---|---|---|
| 1 | 187.89 | 96.65 | 48.53 | 39.94 | 19.84 | 6.64 |
| 2 | 191.19 | 98.24 | 48.88 | 40.72 | 19.91 | 7.29 |
| 3 | 193.84 | 96.78 | 48.73 | 39.68 | 19.48 | 6.75 |
| 4 | 189.55 | 96.28 | 48.72 | 39.88 | 19.50 | 7.22 |
| Average value | 190.62 | 96.99 | 48.72 | 40.01 | 19.68 | 6.98 |

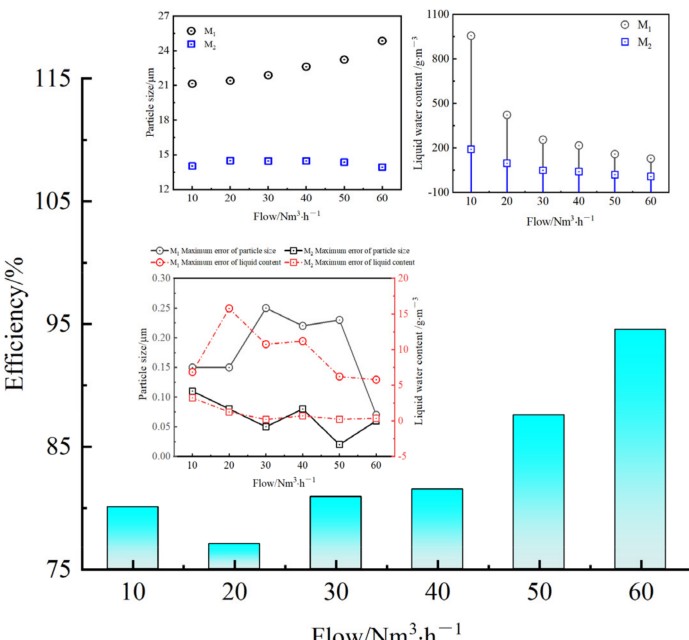

**Figure 6.** Comparison of liquid particle size and content in Case 1 at $M_1$.

The atomized water content was significantly reduced as the air flow increased. The main reason for this phenomenon is because the spray flow rate is constant at 10.44 L/h, and the total flow through $M_1$ rises per unit time when the air flow increases, resulting in the decrease in the relative content of the liquid. After passing through the cyclone element, the liquid particle size decreased from 21.15, 21.40, 22.62, 23.23, 23.23, 24.85 and 24.85 μm to 14.03, 14.49, 14.46, 14.47, 14.36 and 13.93 μm, respectively; the liquid water content decreased from 955.75, 422.26, 255.01, 216.24, 158.05 and 127.55 g/m$^3$ to 190.62, 96.99, 48.72, 40.01, 19.68 and 6.98 g/m$^3$, respectively, indicating that the liquid particle size decreased significantly. Equation (1) was used to calculate the separation efficiency in 10, 20, 30 and 40 Nm$^3$/h flow conditions, which was approximately 80%. Moreover, when the flow rate was higher than 50 Nm$^3$/h, the separation efficiency significantly increased, up to approximately 95% at 60 Nm$^3$/h. The main reason for the increase in the separation efficiency at large flow conditions is that when the flow rate increases, the flow rate of the cyclone element and the intensity of the centrifugal movement increase, resulting in the separation of more droplets. From the experimental results of Case 1, the cyclone element has a good separation effect at a flow rate higher than 50 Nm$^3$/h, but has a lower separation efficiency under small flow conditions.

Additionally, it can be seen from Figure 6 that there are fluctuations in the four measurement results of particle size and liquid content in each working condition. The main reason for the fluctuation is the random error of the experimental measurement, because in the unsteady flow process of the flow field, the flow in the flow field fluctuates. This state is a function of time and the experimental structure is complex. The fluid flow state exhibits turbulent motion and the fluid itself has pulsation values of a random nature. However, from the analysis of the error value, it can be concluded that the measurement error of the experiment was less than 3% (the difference between the average and maximum values of the experimental result was higher than the average value), and the experiment can be considered repeatable.

### 3.2. Studying the Separation Effect of Cyclone Element and Leaf Grid Element Combination

Tables 7–10 show the measurement results of the liquid particle size and content at $M_1$ and $M_2$ under different flow conditions in Case 2. Based on the comparison of Tables 3 and 7 and Tables 4 and 8, due to the differences between cases in the downstream of $M_1$, after changing the combination method, the liquid particle size and content at $M_1$

had minimal impact, and some gaps in the data were caused by uncontrollable factors, such as pipeline vibration during the experiment. After the addition of the leaf grid element, the liquid particle size of each flow condition at $M_2$ was not apparent, but the liquid content in the gas–liquid two-phase flow was reduced further than that in Case 1 (Tables 9 and 10). Figure 7 shows the comparison of the separation efficiency of Cases 1 and 2. The measurement error of the four replicated experiments was within 3%, which is considered reproducible. After increasing the leaf grid element, when the gas flow rate was 10, 20, 30, 40, 50 and 60 $Nm^3/h$, the separation efficiency increased from the previous values of 80.06%, 77.03%, 80.89%, 81.50%, 87.55% and 94.54% to 85.24%, 82.04%, 85.32%, 87.14%, 93.11% and 97.85%, respectively. Therefore, with the addition of the leaf grid element, the separation efficiency of the combined separator was significantly improved when the gas flow rate was lower than 60 $Nm^3/h$. Under the action of strong centrifugation, the separation efficiency of the cyclone element was already high when the flow rate was 60 $Nm^3/h$. Therefore, adding a leaf grid element has no apparent effect on the improvement of the separation efficiency under large flow conditions.

**Table 7.** Liquid particle size in Case 2 at $M_1$ (μm).

| Number of Experiments | 10 $Nm^3/h$ | 20 $Nm^3/h$ | 30 $Nm^3/h$ | 40 $Nm^3/h$ | 50 $Nm^3/h$ | 60 $Nm^3/h$ |
|---|---|---|---|---|---|---|
| 1 | 21.98 | 21.37 | 21.9 | 22.13 | 23.27 | 24.66 |
| 2 | 21.40 | 21.46 | 22.01 | 21.8 | 23.38 | 24.67 |
| 3 | 20.95 | 21.56 | 22.16 | 21.75 | 22.92 | 24.67 |
| 4 | 21.36 | 21.47 | 21.95 | 21.92 | 23 | 24.58 |
| Average value | 21.42 | 21.47 | 22.00 | 21.90 | 23.14 | 24.65 |

**Table 8.** Liquid content in Case 2 at $M_1$ ($g/m^3$).

| Number of Experiments | 10 $Nm^3/h$ | 20 $Nm^3/h$ | 30 $Nm^3/h$ | 40 $Nm^3/h$ | 50 $Nm^3/h$ | 60 $Nm^3/h$ |
|---|---|---|---|---|---|---|
| 1 | 928.74 | 406.47 | 260.94 | 213.95 | 167.14 | 128.94 |
| 2 | 955.75 | 435.11 | 277.50 | 209.79 | 160.67 | 123.94 |
| 3 | 952.69 | 427.70 | 262.58 | 214.11 | 165.74 | 133.40 |
| 4 | 951.62 | 419.74 | 245.34 | 214.52 | 166.27 | 130.76 |
| Average value | 947.2 | 422.255 | 261.59 | 213.0925 | 164.955 | 129.26 |

**Table 9.** Liquid particle size in Case 2 at $M_2$ (μm).

| Number of Experiments | 10 $Nm^3/h$ | 20 $Nm^3/h$ | 30 $Nm^3/h$ | 40 $Nm^3/h$ | 50 $Nm^3/h$ | 60 $Nm^3/h$ |
|---|---|---|---|---|---|---|
| 1 | 13.77 | 14.03 | 14.04 | 14.11 | 14.23 | 13.30 |
| 2 | 13.82 | 13.96 | 14.06 | 14.19 | 14.05 | 13.38 |
| 3 | 13.84 | 14.00 | 14.07 | 14.10 | 14.15 | 13.30 |
| 4 | 13.80 | 14.02 | 14.00 | 14.13 | 14.11 | 13.36 |
| Average value | 13.81 | 14.00 | 14.04 | 14.13 | 14.14 | 13.34 |

**Table 10.** Liquid content in Case 2 at $M_2$ ($g/m^3$).

| Number of Experiments | 10 $Nm^3/h$ | 20 $Nm^3/h$ | 30 $Nm^3/h$ | 40 $Nm^3/h$ | 50 $Nm^3/h$ | 60 $Nm^3/h$ |
|---|---|---|---|---|---|---|
| 1 | 139.77 | 75.26 | 39.14 | 27.39 | 11.72 | 2.86 |
| 2 | 140.21 | 76.86 | 38.42 | 27.68 | 11.61 | 2.74 |
| 3 | 140.41 | 76.63 | 38.16 | 27.2 | 11.51 | 2.87 |
| 4 | 138.84 | 74.57 | 37.93 | 27.38 | 10.62 | 2.65 |
| Average value | 139.81 | 75.83 | 38.41 | 27.4125 | 11.365 | 2.78 |

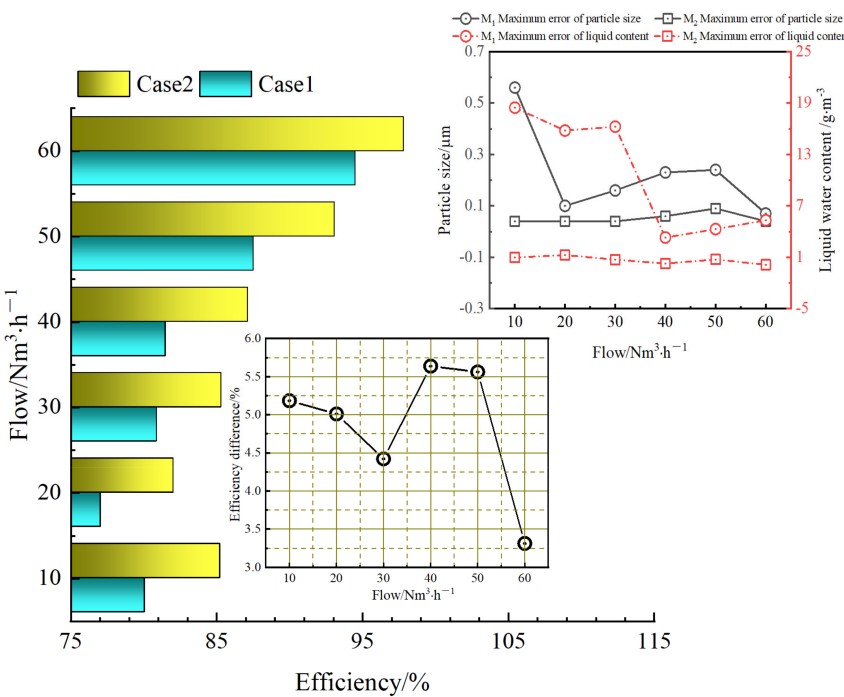

**Figure 7.** Comparison of separation effect between Cases 1 and 2.

### 3.3. Studying the Separation Effect of Cyclone, Leaf Grid and Steady Flow Element Combination

The measurement results of liquid particle size and content at $M_1$ and $M_2$ under different flow conditions in Case 3 are listed in Tables 11–14. In Table 13, after the gas–liquid two-phase flow through the cyclone, leaf grid and steady flow elements, the liquid particle size was approximately 13 μm, which was slightly reduced compared to those of Cases 1 and 2. Based on the data in the table, Figure 8 presents the comparison of the separation efficiencies of Cases 2 and 3, and the experiment can be considered repeatable if the data fluctuation of the four repeated experiments is within 3%. With the addition of the steady flow element, the separation efficiency of each experimental condition was higher than 90% and the separation efficiency increased by nearly 12% compared to that of Case 2 under a 10 $Nm^3$/h flow. The reason for this effect can be explained by the two-phase fluid flow characteristics inside the pipeline. The flow presents a strong nonlinearity when the fluid flows through the cyclone element, whereas the steady flow element is an orifice plate structure. When the turbulent fluid hits the steady flow element, some droplets hang on its surface due to the sudden reduction in the overcurrent section, resulting in a sharp contraction of the fluid volume flowing through the steady flow element. The precipitated part of the liquid remains inside the steady flow element, thus improving the separation efficiency. In a large flow working condition, the cyclone element separates most of the liquid; thus, the separation efficiency of the steady flow element for the large flow working condition does not provide significant improvement. However, the addition of a steady flow element greatly increases the separation efficiency of the combined separator under small flow conditions and enhances the applicability of the combined separator.

**Table 11.** Liquid particle size in Case 3 at $M_1$ (μm).

| Number of Experiments | 10 $Nm^3$/h | 20 $Nm^3$/h | 30 $Nm^3$/h | 40 $Nm^3$/h | 50 $Nm^3$/h | 60 $Nm^3$/h |
|---|---|---|---|---|---|---|
| 1 | 21.47 | 21.69 | 21.98 | 21.41 | 23.16 | 24.59 |
| 2 | 21.72 | 21.36 | 21.92 | 21.47 | 22.93 | 24.81 |
| 3 | 21.99 | 21.58 | 21.97 | 22.12 | 23.28 | 24.59 |
| 4 | 21.72 | 21.42 | 21.83 | 21.84 | 22.94 | 24.63 |
| Average value | 21.73 | 21.51 | 21.93 | 21.71 | 23.08 | 24.66 |

**Table 12.** Liquid content in Case 3 at $M_1$ (g/m$^3$).

| Number of Experiments | 10 Nm$^3$/h | 20 Nm$^3$/h | 30 Nm$^3$/h | 40 Nm$^3$/h | 50 Nm$^3$/h | 60 Nm$^3$/h |
|---|---|---|---|---|---|---|
| 1 | 959.73 | 401.55 | 259.71 | 212.58 | 156.94 | 133.02 |
| 2 | 992.53 | 438.63 | 259.53 | 213.76 | 161.24 | 127.89 |
| 3 | 999.12 | 414.71 | 248.1 | 213.84 | 163.34 | 132.68 |
| 4 | 949.48 | 414.35 | 247.58 | 213.07 | 159.43 | 127.24 |
| Average value | 975.21 | 417.31 | 253.73 | 213.31 | 160.24 | 130.21 |

**Table 13.** Liquid particle size in Case 3 at $M_2$ (μm).

| Number of Experiments | 10 Nm$^3$/h | 20 Nm$^3$/h | 30 Nm$^3$/h | 40 Nm$^3$/h | 50 Nm$^3$/h | 60 Nm$^3$/h |
|---|---|---|---|---|---|---|
| 1 | 13.06 | 13.21 | 13.33 | 13.69 | 13.70 | 13.47 |
| 2 | 13.15 | 13.24 | 13.25 | 13.66 | 13.66 | 13.48 |
| 3 | 13.12 | 13.16 | 13.39 | 13.66 | 13.61 | 13.48 |
| 4 | 13.19 | 13.20 | 13.43 | 13.63 | 13.70 | 13.40 |
| Average value | 13.13 | 13.20 | 13.35 | 13.66 | 13.67 | 13.46 |

**Table 14.** Liquid content in Case 3 at $M_2$ (g/m$^3$).

| Number of Experiments | 10 Nm$^3$/h | 20 Nm$^3$/h | 30 Nm$^3$/h | 40 Nm$^3$/h | 50 Nm$^3$/h | 60 Nm$^3$/h |
|---|---|---|---|---|---|---|
| 1 | 28.02 | 35.35 | 21.56 | 13.78 | 7.42 | 1.90 |
| 2 | 27.27 | 35.17 | 21.30 | 13.89 | 7.55 | 1.93 |
| 3 | 27.25 | 35.56 | 21.12 | 14.08 | 7.49 | 2.00 |
| 4 | 27.62 | 35.43 | 21.68 | 13.68 | 7.44 | 1.89 |
| Average value | 27.54 | 35.38 | 21.42 | 13.86 | 7.48 | 1.93 |

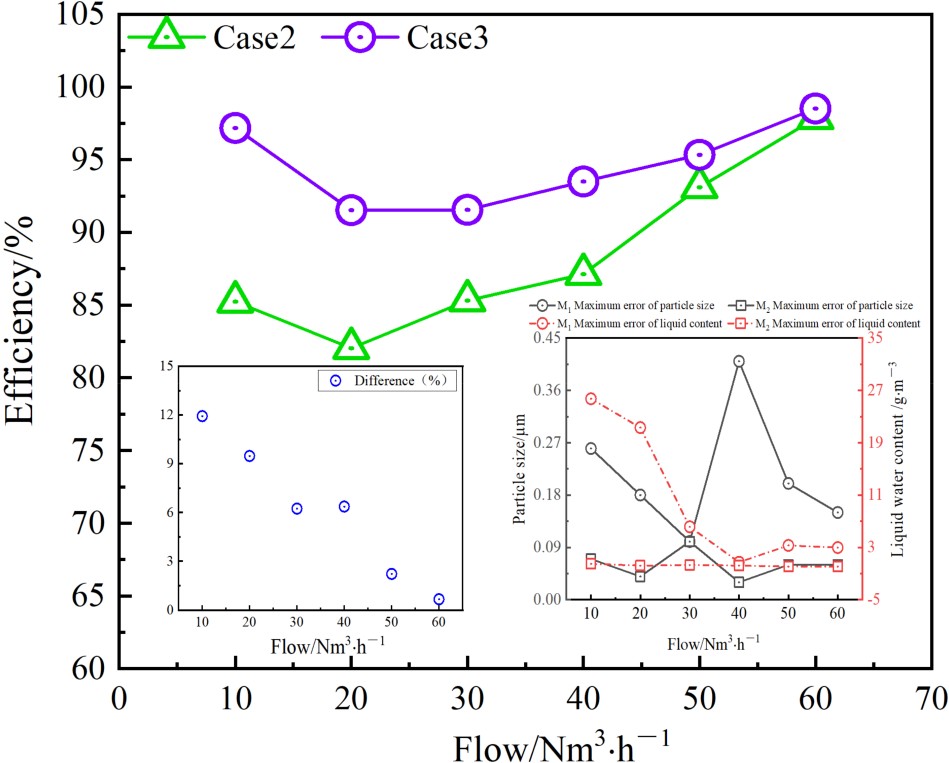

**Figure 8.** Comparison of separation efficiency between Cases 3 and 2.

### 3.4. Studying the Combined Separation Effect of Cyclone, Steady Flow, Leaf Grid and Folding Plate Elements

Tables 15–18 list the measurement results of the liquid particle size and content at $M_1$ and $M_2$ under different flow conditions in Case 4. Based on the data presented in the table, when the gas–liquid two-phase flow passed through the cyclone, steady flow, leaf grid and folding plate elements, the liquid particle size did not change significantly and the particle size under each flow condition was approximately 13 μm. Figure 9 shows the comparison of the efficiency of the four cases. The separation efficiency of Case 4 is higher than that of the other cases, reaching over 95% under each flow condition and as high as 99% at flow rates of 10 and 60 $Nm^3/h$. Due to the addition of the folding plate element, a separation process is added in Case 4, but this only improves the separation efficiency by 0.55% compared to that of Case 3 at the flow condition of 60 $Nm^3/h$; thus, its effect is not apparent. In large flow working conditions, the centrifugal separation generated by the cyclone element increased the separation efficiency to 94.54%. Combined with the cyclone, steady flow and leaf grid elements, the liquid separation effect reached over 98%. Thus, with the addition of the folding plate element, the separation efficiency is somewhat increased. However, when the flow rate is less than 60 $Nm^3/h$, due to the low degree of centrifugation of the cyclone element, the moisture cannot be fully removed. Downstream of the cyclone element, the dehydration effect of the steady flow and leaf grid elements is slightly reduced due to the small turbulence intensity of the gas–liquid two-phase flow. Therefore, this new separation process must be added to further improve the separation efficiency. The results show that under a flow condition of approximately 10–50 $Nm^3/h$, the separation efficiency increased by more than 2% due to the addition of a folding plate element, which provides optimal conditions for efficient separation.

**Table 15.** Liquid particle size in Case 4 at $M_1$ (μm).

| Number of Experiments | 10 $Nm^3/h$ | 20 $Nm^3/h$ | 30 $Nm^3/h$ | 40 $Nm^3/h$ | 50 $Nm^3/h$ | 60 $Nm^3/h$ |
|---|---|---|---|---|---|---|
| 1 | 21.12 | 21.24 | 22.19 | 21.32 | 23.23 | 24.80 |
| 2 | 21.56 | 20.94 | 22.02 | 21.8 | 23.08 | 24.66 |
| 3 | 21.72 | 21.15 | 21.72 | 21.62 | 23.38 | 25.00 |
| 4 | 21.86 | 21.3 | 21.8 | 21.87 | 23.15 | 24.95 |
| Average value | 21.57 | 21.16 | 21.93 | 21.65 | 23.21 | 24.85 |

**Table 16.** Liquid content in Case 4 at $M_1$ ($g/m^3$).

| Number of Experiments | 10 $Nm^3/h$ | 20 $Nm^3/h$ | 30 $Nm^3/h$ | 40 $Nm^3/h$ | 50 $Nm^3/h$ | 60 $Nm^3/h$ |
|---|---|---|---|---|---|---|
| 1 | 910.25 | 432.61 | 245.39 | 217.66 | 163.86 | 131.1 |
| 2 | 988.53 | 422.17 | 272.76 | 202.71 | 168.93 | 131.23 |
| 3 | 932.43 | 405.27 | 253.41 | 206.68 | 163.37 | 133.3 |
| 4 | 990.02 | 433.8 | 250.74 | 209.74 | 160.74 | 132.36 |
| Average value | 955.31 | 423.46 | 255.58 | 209.20 | 164.22 | 132.00 |

**Table 17.** Liquid particle size in Case 4 at $M_2$ (μm).

| Number of Experiments | 10 $Nm^3/h$ | 20 $Nm^3/h$ | 30 $Nm^3/h$ | 40 $Nm^3/h$ | 50 $Nm^3/h$ | 60 $Nm^3/h$ |
|---|---|---|---|---|---|---|
| 1 | 12.73 | 13.34 | 13.8 | 13.91 | 13.99 | 13.75 |
| 2 | 12.83 | 13.37 | 13.81 | 14.02 | 13.98 | 13.77 |
| 3 | 12.68 | 13.31 | 13.75 | 13.98 | 13.95 | 13.68 |
| 4 | 12.67 | 13.37 | 13.37 | 13.92 | 13.87 | 13.84 |
| Average value | 12.73 | 13.35 | 13.68 | 13.96 | 13.95 | 13.76 |

**Table 18.** Liquid content in Case 4 at $M_2$ (g/m³).

| Number of Experiments | 10 Nm³/h | 20 Nm³/h | 30 Nm³/h | 40 Nm³/h | 50 Nm³/h | 60 Nm³/h |
|---|---|---|---|---|---|---|
| 1 | 8.75 | 15.35 | 12.23 | 8.62 | 3.81 | 1.19 |
| 2 | 7.38 | 15.41 | 12.56 | 8.27 | 3.83 | 1.30 |
| 3 | 5.86 | 15.14 | 12.18 | 8.22 | 4.34 | 1.25 |
| 4 | 5.17 | 15.15 | 12.55 | 8.40 | 4.11 | 1.18 |
| Average value | 6.79 | 15.26 | 12.38 | 8.38 | 4.02 | 1.23 |

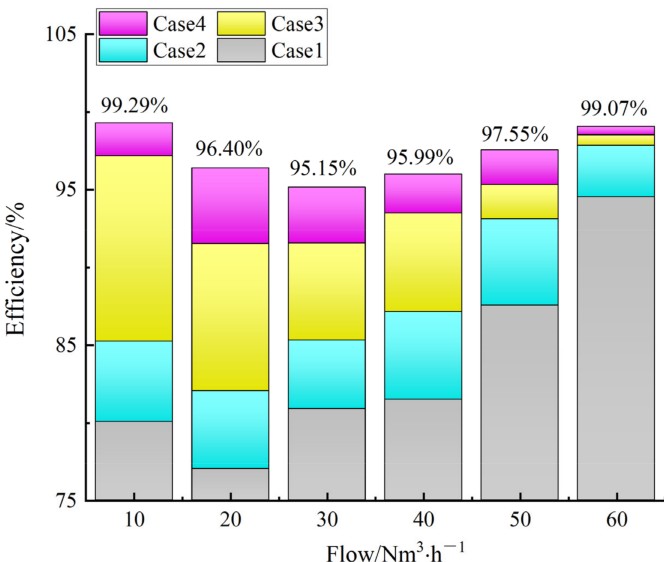

**Figure 9.** Comparison of the separation efficiency in the four cases.

Table 19 shows a comparison between the combined separator and traditional separators. While the separation efficiency of the combined separator is similar to that of the other separators, the combined separator has a wider range of applicability (large and small flows have higher separation efficiencies) and occupies a smaller space. Figure 10 shows the error plot of the four replications in Case 4, and the error result is still within 3%, which verifies that the reproducibility of the experiment is high.

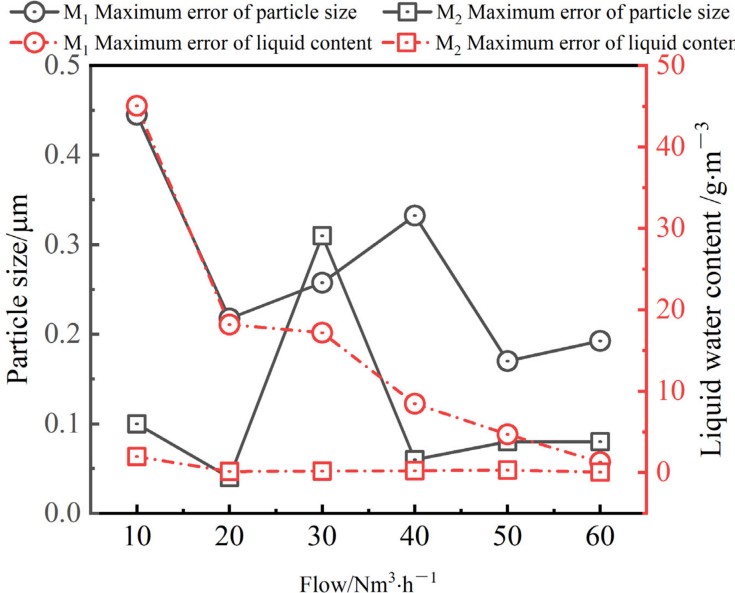

**Figure 10.** Error diagram for four repetitions.

**Table 19.** Comparative analysis of the applicability of all separators.

| Name | Suitable Conditions | Separation Efficiency |
|---|---|---|
| Gravity separator | Gas–liquid separation of droplet size from 60 to 100 μm | Low efficiency, generally used for primary separation |
| Multi-tubular cyclone separator | Heavy loads | 40–50 μm droplets: >98%<br>5–10 μm droplets: >90% |
| Inertial gas–liquid separator | Large particle sizes | Low when particle size is <25 μm |
| Combined separator | Wide application range (suitable for large and small flow rates) | >95% |

## 4. Conclusions

In this study, a new combined separator was designed. The separation assembly includes the cyclone, steady flow, leaf grid and folding plate elements, and the experimental study of different combinations at approximately 10–60 $Nm^3$/h flow conditions was conducted using LDA and PDPA equipment. The main conclusions of this study are as follows.

(1) When the combined separator only relied on the cyclone element for gas–liquid separation, the separation efficiency was approximately 80% at a flow rate of 10–50 $Nm^3$/h and approximately 95% at a flow rate of 60 $Nm^3$/h. Thus, this method is only suitable for large flow conditions and its applicability is low.

(2) The gas–liquid separation efficiency under the flow conditions of 10, 20, 30, 40 and 50 $Nm^3$/h increased by 5.18%, 5.01%, 4.43%, 5.64% and 5.56%, respectively, when the combined mode comprised the cyclone and leaf grid elements. Therefore, the addition of a leaf grid component significantly improves the efficiency of the combined separator under small flow conditions.

(3) When the combined separator comprised the cyclone, steady flow and leaf grid elements, the separation efficiency increased by approximately 12% at a flow rate of 10 $Nm^3$/h. Moreover, the separation efficiency under the working conditions of 20, 30, 40, 50 and 60 $Nm^3$/h also significantly increased. The separation efficiency of the combined separator was higher than 90% at flow rates of 10–60 $Nm^3$/h, which enhances the applicability of the combined separator.

(4) When the combined separator included the cyclone, steady flow, leaf grid and folding plate elements, the separation efficiency was higher than 95% when the flow rate was in the 10–60 $Nm^3$/h range, and the separation efficiency exceeded 99% at flow rates of 10 and 60 $Nm^3$/h, indicating that the separator has an efficient separation effect.

**Author Contributions:** Conceptualization, Methodology and Software, L.J.; Data Curation, Writing and Original Draft Preparation, Q.Z.; Visualization and Investigation, H.D.; Supervision, L.Z.; Software and Validation, W.D. All authors have read and agreed to the published version of the manuscript.

**Funding:** This research was funded by the National Key Research and Development Program "Research and Application Demonstration of Complementary Combined Power Generation Technology between Distributed Photovoltaic and Cascade Small Hydropower" (Grant No. 2018YFB0905200); Science and technology research project of Education Department of Jiangxi Province (Grant No. GJJ211941).

**Data Availability Statement:** Not applicable.

**Acknowledgments:** The authors express their gratitude to Deng for providing language help and for his assistance during the experiments.

**Conflicts of Interest:** The authors do not have any conflict of interest to declare.

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
