# Peer review of "Experimental Study on a New Combined Gas–Liquid Separator"

_processes, doi:10.3390/pr10071416_

Round 1
Reviewer 1 Report
I am recommending publication of the article in its current form.
Author Response
Thank you for your comments and support

Reviewer 2 Report
The experimental study on a new combined gas-liquid separator is well studied. This manuscript can be accepted after the minor revision. Please find my minor comments below.
1. There was not any error analysis performed throughout the study. To be reproducible, it is very important to have the error bars on all the figures.
2. Figure 1 caption is not more informative.
3. Please maintain uniformity. For eg, in some places, it is mentioned as Figure, and in some places, it is mentioned as Fig.
4. In Figure 8, please write as Efficiency (%) and Flow (gm-3). Please correct the same everywhere in the manuscript.
5. The obtained result should be compared with the literature and tabulated. It will be very helpful to assess the importance of the present study.
6. In conclusion, please correct it as Nm3/h (line number 244).
Author Response
Revised draft reply
Dear reviewers,
I received your suggestion on revision on June 29, 2022, and read it carefully. After serious reflection, I found that the questions you raised were details that I did not take into account in manuscript writing. I am very grateful for your precious time to help me improve the quality of the manuscript. I made the following modifications according to your suggestions:
- "scheme" in the text and "case" in the figures;
Thanks for the expert's valuable advice. Because of my carelessness, I have replaced all schemes in the article with cases, please check.
- In Figure 1 and in Figure 2, there are other component names than in the scheme, it is not clear where the cyclone is in the scheme.
The components (Figure 1) should be described in more detail (how they were designed, their configurations, dimensions, etc.).
Thanks to the valuable advice of the experts, the Whirlwind compoents location is shown in the figure. He is located at the lower end of Observation Point 1, but I can't provide the exact values for now because the layout of the school laboratory has changed and the experimental bench has been demolished. Since the project is jointly completed by us and petroChina Southwest Oil and Gas Field Research Institute, we provide experimental conditions and technology, they design various components, and the design process and dimensional parameters are confidential, so we can not provide, please understand! We're sorry!
Thank you again for the help and support of experts, through your comments on the manuscript, I understand your rigorous academic attitude, in the future study, I will work harder, to your example!
